# Modifying the Expression of Cysteine Protease Gene *PCP* Affects Pollen Development, Germination and Plant Drought Tolerance in Maize

**DOI:** 10.3390/ijms24087406

**Published:** 2023-04-17

**Authors:** Yanhua Li, Liangjie Niu, Xiaoli Zhou, Hui Liu, Fuju Tai, Wei Wang

**Affiliations:** National Key Laboratory of Wheat and Maize Crop Science, College of Life Sciences, Henan Agricultural University, Zhengzhou 450046, China; lyhyh908@outlook.com (Y.L.); niuliangjie@aliyun.com (L.N.);

**Keywords:** pollen cysteine protease (PCP), pollen germination, drought tolerance, transgenic plants, *Zea mays*

## Abstract

Cysteine proteases (CPs) are vital proteolytic enzymes that play critical roles in various plant processes. However, the particular functions of CPs in maize remain largely unknown. We recently identified a pollen-specific CP (named PCP), which highly accumulated on the surface of maize pollen. Here, we reported that PCP played an important role in pollen germination and drought response in maize. Overexpression of *PCP* inhibited pollen germination, while mutation of *PCP* promoted pollen germination to some extent. Furthermore, we observed that germinal apertures of pollen grains in the *PCP*-overexpression transgenic lines were excessively covered, whereas this phenomenon was not observed in the wild type (WT), suggesting that PCP regulated pollen germination by affecting the germinal aperture structure. In addition, overexpression of *PCP* enhanced drought tolerance in maize plants, along with the increased activities of the antioxidant enzymes and the decreased numbers of the root cortical cells. Conversely, mutation of *PCP* significantly impaired drought tolerance. These results may aid in clarifying the precise functions of CPs in maize and contribute to the development of drought-tolerant maize materials.

## 1. Introduction

The papain-like cysteine protease (CP) is a crucial endopeptidase belonging to a large family found in most living organisms, and playing a significant role in protein degradation and programmed cell death (PCD) [1]. PCD is a process of cellular self-extinction that occurs in response to specific gene expression changes, such as those involving cysteine proteases or senescence-specific cysteine proteases. In plants, papain-like CPs are typically active in acidic subcellular compartments, such as cell walls (apoplasts), lysosomes and vacuoles. They play a positive role in diverse PCD events during organ differentiation, aging tissues, xylem initiation, lateral root formation, leaf and flower senescence, pollen development, seed germination and seedling development [2,3,4].

CP is first synthesized as proenzymes within precursor protease vesicles, which contain a C-terminal KDEL endoplasmic reticulum (ER) retention signal. These vesicles are then transported to the vacuole. Upon removal of the KDEL tail, the enzyme is activated and converted into its mature form before vacuole breakdown [2,5]. KDEL-tailed CP is the only protease capable of digesting extensins that form cell walls and basic scaffolds [2]. The expression of *CP* is promoter- and tissue-specific during seedling, flower and root development. For example, in *Arabidopsis*, a *CP* (*CEP1*) was specifically expressed in the tapetum at the 5th to 11th stages of anther development [5]; in maize, a CP is synthesized during stage four of developing anthers [6].

In angiosperms, tapetum programmed cell death (PCD) is a critical prerequisite for pollen development. To date, many CPs have been implicated in tapetum PCD and pollen development in various species, including *Arabidopsis* [5,7], *Nicotiana tabacum* [8] and *Oryza sativa* [9]. In *Arabidopsis*, the loss of *CP* (*cep1*) function in the anther tapetum results in abnormal pollen exine and decreased pollen fertility, whereas overexpression of *CEP1* leads to premature tapetal PCD and pollen sterility [5]. *CEP1* serves as the key executor of tapetal PCD, and its proper expression is necessary for the timely degeneration of tapetal cells and the formation of functional pollen. Similarly, in *Arabidopsis*, *AtCP51* is essential for proper pollen exine, and reduced expression levels result in premature degradation of the tapetum and defective exine of pollen [7]. In rice, *OsCP1*, strongly expressed in anthers, was silenced by T-DNA insertional mutagenesis, leading to pollen degeneration [9]. *NtCP56* is highly expressed in tobacco anthers, and knockdown of *NtCP56* leads to the retardation of tapetum degradation and the production of aborted pollen [8]. These findings suggest that CPs play a crucial role in tapetum PCD and pollen development across plant species.

In the plant sexual reproductive process, sexual reproduction begins when pollen grains come into contact with the stigma of a flower [10]. Maize is an economically important model organism commonly used for studying wind pollination and fertilization mechanisms [11,12]. In a previous study, we extracted and purified the surface components of maize B73 pollen grains using chloroform/phenol and identified a set of 12 major pollen surface proteins via proteomic approach. Among these proteins, PCP was found to exist in high abundance on the pollen surface [12]. We further localized PCP enrichment in the inner wall layer of pollen and in a small quantity within pollen cytoplasm via immunofluorescence microscopy and immunogold electron microscopy [13]. Notably, the proteomic composition of maize pollen surfaces differed significantly from that identified in *Arabidopsis* [14]. Maize PCP is the only known tapetum protease that appears at the late stage of anther development in any species [6]. No analogous PCP has been identified on the pollen surface of other cereal plants or dicots, such as *Arabidopsis* [14]. It is possible that maize CP is involved in tapetum PCD [6] and plays a role during pollen development and germination [15].

Maize PCP belongs to the protease C1 family, with its encoding gene (*PCP*, LOC 100280441) measuring 1052 bp in length and located on the first chromosome (75210848-75212440; single copy). Bioinformatic analysis has shown that maize PCP is relatively conserved during evolution, with low homology with other maize CPs (≤39%) [13]. Like other maize pollen surface enzymes, such as endoxylanase and glucanase [11,16], maize PCP also originates from the tapetum. It is initially synthesized as inactive precursors in the endoplasmic reticulum (ER), then transported and stored in vacuoles. As tapetal cells approach PCD, PCP is released onto the pollen surface [6]. Maize *PCP* is preferentially expressed in anthers and pollen grains but is expressed only lowly in other tissues, such as roots, stems and leaves [13]. Although many studies have demonstrated that anther CPs are involved in pollen development, the function of maize PCP in pollen germination and tube growth remains largely unclear.

CP also plays a crucial role in defending plants against abiotic stresses, particularly water stress (drought, waterlogging) [17]. For instance, the expression of sweet potato *SPCP3* in *Arabidopsis* led to increased drought sensitivity, while ectopic expression of *SPCP2* conferred drought resistance [18]. Overexpression of *TaCP* from *Triticum aestivum* in *Arabidopsis* enhanced the drought resistance of transgenic plants [19]. The expression levels of *TaWCP2* in wheat were inhibited in cultivars with high drought tolerance, while remaining unchanged or increasing in cultivars with low drought tolerance [20]. The expression of sweet potato *CP2* altered the developmental and stress response characteristics of transgenic *Arabidopsis* plants [18]. However, *Carthamus tinctorius CP1* acts as a negative regulator in the response of transgenic *Arabidopsis* to low-temperature stress; that is, inhibiting the expression of *CtCP1* enhanced plant cold resistance [21]. Recently, we identified the existence of defense and stress response elements (e.g., TC-rich repeats) and MYB binding sites in maize *PCP* (designated as *ZmCP03*), implying a possible role in drought resistance [13].

The aim of this study was to investigate the role of PCP in maize pollen germination and drought tolerance. We analyzed the effects of *PCP* expression levels on pollen morphology, structure, viability, in vitro germination and drought tolerance in transgenic plants by employing *pcp* knock-out (*KO*) and overexpressed (*OE*) maize lines.

## 2. Results

### 2.1. Identification of the pcp Mutants

To investigate the role of PCP during maize pollen development, we utilized the CRISPR/Cas9-based system to generate *pcp* mutant lines. The T0 mutant plants that exhibited Basta resistance were self-pollinated to obtain T1 seeds, which were germinated in Basta-containing medium and subsequently transplanted into soil to produce homogenous T2 seeds. To identify homozygous *pcp* lines, sgRNA target sites in T1 (leaf) mutant plants were sequenced. Amplification of a 633 bp fragment using PCP-F2/R2 primers revealed the presence of three target sites with single peaks only (Figure 1), indicating the homozygous nature of these mutants.

Several mutations were detected in the pollen grains of T2 mutants. Among them, three homozygous mutants (designated *KO1*, *KO2* and *KO3*) were selected for further investigation. Comparative analysis of these mutants with the reference sequence revealed that *KO1* had an additional 27 bp sequence and one less ‘T’ base at target 2, while no mutations were present in targets 1 and 3. Conversely, *KO2* exhibited a 23 bp deletion between targets 1 and 2, and a missing ‘C’ base in target 3. *KO3* had 1, 4 and 5 bp absent at targets 1, 2 and 3, respectively (Figure 1). Sequencing data of PCR products demonstrated single peaks only, indicating the homozygosity of *pcp* in the mutants (Appendix A).

To further ascertain the expression of *PCP* in mutant pollen, RNA was extracted from pollen grains of T3 mutants and subjected to reverse transcription quantitative PCR (RT-qPCR). As depicted in Appendix A, the expression level of the *PCP* gene considerably declined in the mutant pollen.

### 2.2. Generation of PCP Overexpressed Lines

An overexpression vector of *PCP* under the control of the cauliflower mosaic virus 35S promoter was constructed. By employing the agrobacterium-mediated method, 30 *PCP* overexpressed maize plants were obtained (Figure 2A). Among them, the relative expression level of *PCP* in leaves at the three leaf stage of plants 2, 3, 8–10, 12, 14, 17–19, 25 and 30 under laboratory conditions was significantly higher than that of WT, accounting for 43% of the total plants. The seeds of 12 and 30 T0 overexpression plants were grown in the field to produce stable genetic overexpression transgenic plants (Figure 2B). Interestingly, the expression level of *PCP* in leaves at anthesis stage of plant 12 and 30 under field conditions was notably higher than that of WT. Furthermore, overexpression of *PCP* in pollen was confirmed in T2 *PCP-OE-12* and *PCP-OE-30* each using five randomly selected plants (Figure 2C). Subsequently, *PCP-OE-12* and *PCP-OE-30* lines were selected for further analysis, designated as *OE1* and *OE2*, respectively.

### 2.3. Pollen Morphology and Structure Comparison

Scanning electron microscopy (SEM) was employed to analyze the morphology of pollen grains in different maize plants. Results revealed that the apertures of *OE1* and *OE2* pollen grains were covered by a convex operculum, unlike those of WT, *KO1* and *KO2* pollen grains that were covered by an invaginated operculum (Figure 3). This invaginated operculum is typically observed in normal maize [22] and other *Gramineae* plants [23]. Additionally, the surface of *OE* pollen grains appeared relatively smooth compared to the *pcp* and WT pollen grains with an evident concave–convex ornament (Figure 3). However, no significant changes were observed in the morphology and size of these pollen grains.

To compare the internal structures of pollen grains from three genotypes, transmission electron microscopy (TEM) was employed (Figure 4). The surface attachment of pollen exine was significantly increased in *KO* pollen, followed by WT, and almost disappeared in *OE* pollen (Figure 3 and Figure 4). Moreover, an abnormally high accumulation of starch granules was observed in some *KO2* pollen grains (Figure 4).

### 2.4. In Vitro Pollen Viability and Pollen Germination

Most pollen grains of *PCP-OE* lines exhibited low viability, as they appeared unstained or stained as light red by 2,3,5-triphenyltetrazolium chloride (TTC) staining (Figure 5). In contrast, pollen grains of *pcp* mutants (*KO1* and *KO2*) and WT were mostly red-stained, indicating higher viability rates (Figure 5). The relative pollen viability between WT and *OE* lines was significantly different, while no significant difference was observed between WT and *KO* lines (Figure 5) based on statistical analysis across the three pollen genotypes using I_2_-KI staining.

To evaluate the actual pollen viability, in vitro pollen germination and tube growth were compared across WT and transgenic lines. A significant difference in germination was observed between WT and *OE* pollen grains (Figure 6, Appendix A). In the presence or absence of the CP inhibitor E64, WT pollen germination remained similar but was significantly higher than *OE* pollen grains, which showed a reduced germination rate. Interestingly, the germination rate and tube growth of *OE* pollen grains were increased to some extent by adding E64, especially after 4 h germination (Figure 6, Appendix A). These results indicated that PCP may play a negative regulatory role during maize pollen germination. In the presence of E64, the percentage of pollen germination rate was higher in *KO1* and *KO2* than in WT (Figure 6A). However, the length of pollen tube was reduced in *KO2* with or without E64 in comparison to WT (Figure 6B). Thus, *pcp* mutation exerted a certain impact on pollen germination in vitro.

### 2.5. Comparison of Plant Phenotypes among WT and Transgenic Maize

To investigate the effect of *PCP* expression level on pollination, seed setting rate was compared among WT and transgenic plants under field conditions (Figure 7, Appendix A). The seed setting rate was considerably lower in *OE* plants and *KO1* than in WT (Figure 7B), potentially correlating with the performance of pollen germination and tube growth in different genotypes under field conditions. Moreover, the knockout or overexpression of *PCP* had no visible effect on plant height (Figure 7C). The number of tassel branches increased in *OE1* plants, whereas it decreased in *OE2* plants (Figure 7D). This difference may be attributed to individual differences between *OE1* and *OE2*.

### 2.6. Comparison of Drought Tolerance between WT Maize and Transgenic Lines

After 14 d of soil drought treatment, *pcp* mutant plants exhibited drought sensitivity, with early and severe leaf wilting compared to WT plants (Figure 8A). After 3 d of rehydration, the survival rate of *KO* plants was only 20%, significantly lower than the 80% observed in WT plants.

After 17 d of soil drought treatment, all leaves of WT plants exhibited serious withering (Figure 8A), while only the first leaf wilted in *OE* plants. After 3 d of rehydration, the survival rate of *OE* plants was above 90%, whereas it was only around 40% in WT plants. Thus, *OE* plants demonstrated higher drought resistance.

Next, physiological and biochemical indices related to drought tolerance were compared among WT and transgenic lines. After 7 d of drought treatment, the relative water content (RWC) of leaves decreased significantly, especially in the *KO1* line. Furthermore, the malondialdehyde (MDA) content in the *KO1* line was significantly higher than that in WT and *OE1* plants (Figure 8B). The glutathione (GSH) contents increased significantly in *OE1* plants. Under drought stress, the content of reactive oxygen species (ROS) (e.g., O_2_^−^, H_2_O_2_) increased significantly in the leaves of WT, *KO1* and *OE1* plants. The activities of antioxidant enzymes (SOD, POD, APX, GR, DHAR, MDHAR) were much higher in the leaves of *OE1* plants than in those of WT and *KO1* plants (Figure 8B, Appendix A). These results indicate that *KO* plants possess weak anti-oxidation machinery compared to WT and OE plants, resulting in inefficient scavenging concerning the accumulation of intracellular ROS.

To investigate the role of *PCP* in root response to drought stress, RT-qPCR was performed to quantify its expression levels. The results showed that *PCP* was highly expressed in the root, and its levels were significantly enhanced by drought stress (Figure 9A). The changes in root microstructure among WT and transgenic plants were compared to examine the effect of *PCP* expression levels (Figure 9A). The root diameter of *OE* plants was significantly smaller compared to WT and *KO* plants, primarily due to the lower layer numbers of cortex cells (plus autophagy during root development under normal and drought stress conditions). Accordingly, the proportion of central stele (vascular tissues) decreased in *OE* roots, along with an altered arrangement in vessels, compared to *KO* and WT roots (Figure 9B,C).

## 3. Discussion

Several prevalent pollen surface proteins, including PCP, have been suggested to play possible roles in pollen development and tube growth in maize [6,12,13,24,25]. As the only known protease identified on the maize pollen surface, however, the direct evidence regarding the function of PCP has remained elusive. In the present study, we created *pcp* mutants and overexpressed *PCP* lines to investigate the role of PCP in both pollen germination and drought tolerance in maize.

### 3.1. The Increased Expression of PCP Negatively Affects Maize Pollen Germination

Differences in pollen surface morphology can influence the total amount of pollen coat and thus impact pollen grain transmission and interaction with stigmas [10]. Additionally, differences in pollen wall morphology can affect pollen hydrodynamics, which refers to the pollen grains’ ability to change volume due to changes in water content without loss of viability [26].

Germination apertures are the sites where pollen tube growth is initiated. In nature, the morphology of pollen apertures varies and plays a crucial role in pollen germination, particularly in facilitating water uptake and expulsion through these apertures to regulate pollen volume and the rate of water entry upon pollen hydration [27,28]. The loss of apertures has been shown to affect pollen germination rates in *Arabidopsis* [29], rice [28] and maize [30]. In rice, the loss of *OsDAF1* resulted in the abortion of most pollen, and the surviving pollen lacked germination pores and failed to germinate both in vitro and in vivo; similarly, *OsINP1* functions in the late formation of germination pores, possibly by preventing the deposition of outer wall precursors [28]. Aperture number may also impact pollen grain performance. For instance, in *Arabidopsis*, triaperturate pollen grains perform better than those with other aperture numbers [31].

The surface morphology of pollen grains has been linked to pollen viability. The rough or smooth surface of maize pollen grains can affect their adhesion and hydration on the stigma surface, resulting in further pollen germination and kernel setting [32]. A study of rice *OsCNGC13* found that the gene encoding this protein, *sss1-d*, had a reduced kernel setting rate when knocked down, and cellular observation revealed that approximately half of the mutant pollen tubes were stunted in the stigma [33]. Transcriptome analysis showed that the expression of 872 genes in *cep1* mutants changed significantly, most of which were important for the formation of tapetal cell wall tissue, tapetal secretory structure and pollen development. *CEP1* expression levels indicated that the degree of tapetal PCD was closely related to pollen fertility [5].

Our study revealed that the surface morphology of *OE* pollen grains was smoother than that of *KO* and *WT* grains, which may affect pollen–stigma interaction and further pollen germination. Operculum is commonly observed in *Gramineae* pollen grains that all have ulcerate apertures [28,34]. It is an exine thickening that covers most of the aperture, especially in wind-pollinated maize. Its function is likely to protect the cytoplasm from desiccation and prevent pathogen entry via the aperture [23]. However, the aperture of *OE* pollen was excessively covered by a thick convex operculum, which might affect the pollen’s timely perception of the change in external moisture and further initiation of germination. We speculate that PCP may be related to the deposition of outer wall precursors. The presence of convex opercula may be one reason for the decreased pollen germination rate and decreased kernel setting rate observed in *OE* lines. Starch biosynthesis is crucial for pollen viability and maturation in cereal crops such as maize [35] and rice [36]. We also observed the high accumulation of starch granules in *pcp* pollen grains, although the mechanism governing this process remains largely unknown. Based on our pollen viability, surface structure, germination and kernel setting results, we suggest that the expression levels of PCP are negatively correlated with pollen germination.

In addition, in this study, *KO2* pollen length was significantly lower than that of *KO1* plants incubated in vitro for 4 h. The variance in pollen tube length between *KO1* and *KO2* may be caused by their different mutation locations (Figure 1) and *PCP* expression levels (Appendix A). Similar phenomena have been observed in maize [37] and *Setaria viridis* [38] *KO* mutants. For instance, internode lengths and numbers were clearly different in the *ZmGA20ox3* gene *KO* mutants [37]. A single codon deletion mutant and two *ANT1* orthologous frameshift mutants of *Setaria viridis* generated through CRISPR/Cas9 technology displayed noticeable variations in photosynthesis efficiency, growth rate, leaf size and grain yield [38]. Moreover, in this study, the variation in the number of tassel branches between *OE1* and *OE2* plants might be attributable to the differential levels of expression of the *PCP* gene. Individual *OE*-lines of maize (e.g., *YIGE1-OE*, *ZmPHYC1-OE*) showed differences to some extent in ear length, grain yield [39] and organ growth [40]. The molecular mechanisms underlying these phenotype differences are largely unclear.

### 3.2. The Increased Expression of PCP Positively Enhances Maize Drought Tolerance

Certain CPs have been implicated in plant responses to various abiotic and biotic stresses, including drought, salt and pathogen infection [41,42]. The upstream region of the *PCP* sequence contains MYB binding sites and ABA responsive elements (ABREs) that are related to drought induction [13]. In this study, we compared the drought responses of maize seedlings and found that the *OE* lines were more drought-resistant, followed by WT, and the *pcp* mutants were the most susceptible to drought. This could be attributed to the potent antioxidant machinery in *PCP-OE* maize plants, as enhancing cellular redox homeostasis is crucial for plants to tolerate drought stress [43,44]. Compared to WT and *pcp* mutants, *PCP-OE* plants exhibited high abilities to cope with ROS-induced oxidative damage, as evidenced by decreased MDA content, increased GSH content and increased antioxidant enzyme activities in leaves under drought stress.

The root is the primary plant organ responsible for detecting changes in soil conditions and plays a critical role in water stress response [45,46,47]. The molecular mechanism of drought resistance through roots primarily includes the regulation of ABA synthesis in roots to regulate the expression of genes involved in drought signal transduction [48]. Plants typically allocate relatively more resources to the root system in response to drought stress [49]. Maize root responses to drought stress depend on root class and axial position [50]. The formation of more cortical ventilation tissue in maize roots also enhances drought resistance by decreasing root respiration and increasing root depth, thus improving leaf water status [49,51]. Reduced cortex number has been associated with reduced root respiration per root length, whereas under water stress conditions, lines with reduced cortex number had longer roots and increased stomatal conductance, leaf CO_2_ assimilation and aboveground biomass compared to lines with higher cortex number [52,53]. Nutrient deficiency (N, P and K) can strongly modulate exodermal differentiation in maize roots [54].

In the present study, we found that the number of cortex cells in the roots of *PCP-OE* lines was significantly decreased when compared to WT and *KO* plants at 30–40 mm from the root tips. Additionally, the aerenchyma formed in grasses is a lysogenic aerenchyma, and PCD initiated by CP may play a role in aerenchyma formation in maize [55]. After drought stress, the expression of *PCP* in the roots increased gradually with stress time. It is speculated that the overexpression of *PCP* might induce PCD of root cortex cells, reducing root metabolic attrition in response to drought stress.

## 4. Materials and Methods

### 4.1. Plant Materials, Growth Conditions and Phenotyping

Maize (*Zea mays*) inbred line B104 was used for all gene transfer experiments and as the wild-type (WT) control. Plants were grown under normal field conditions at Henan Agricultural University’s experimental farm in Zhengzhou, China, over three consecutive planting seasons from May to September in 2020–2022. During anthesis, fresh pollen grains were collected from the tassels by shaking them into paper bags between 9:00 a.m. and 11:00 a.m. The pollen grains were then purified to remove any impurities and used for viability and germination tests or dehydrated over silica gel for 24 h and stored at −20 °C until use. Plant phenotypes were analyzed via digital camera photography, and kernel setting rates were counted during harvest time.

### 4.2. Drought Treatment

Soil drought treatment of maize plants was conducted according to a previously described method [56]. Briefly, maize seeds were germinated in soil and watered every 3 d. The seedlings were grown under controlled conditions of 27/22 °C (day/night) and 60% relative humidity with a 14 h light/10 h dark cycle and a light intensity of 500 mmol m^−2^ s^−1^. At three leaf stage, watering was stopped until the soil became dry (approximately 14 d). The phenotypes of maize plants were recorded and repeated after 3 d of rehydration. After seven days of drought treatment, the 2nd and 3rd leaves were separately sampled for physiological and biochemical assays.

For PEG-6000 osmotic stress treatment, maize seedlings were cultivated at 27 °C under 500 mmol m^−2^ s^−1^ light for 14 h per day. At three leaf stage, the seedlings were immersed in a 10% PEG-6000 + 1/4 strength Hoagland’s solution for 0, 3, 6 and 12 h, respectively. Roots and leaves were sampled and directly used for physiological and biochemical assays.

### 4.3. Generation of PCP Overexpressed Lines

For overexpression vector construction, the CDS of *PCP* was amplified from maize genomic DNA with the PCP-F/R primer pair and inserted in the binary vector pBWA(V)BS that was driven by the cauliflower mosaic virus 35S promoter (Appendix A). The pBWA(V)BS-*PCP* vector was introduced into maize embryo cells via *Agrobacterium tumefaciens* (strain GV3101) mediated transformation [57]. The T0 generation *PCP* overexpression (*OE*) lines were obtained through Basta-resistance selection at 100 mg/L, confirmed via RT-qPCR, and planted in the field for self-pollination to produce seeds.

### 4.4. Generation of CRISPR-Cas9 Mutants

To generate CRISPR-Cas9 mutants via mutagenesis of *PCP* (GenBank accession: LOC100280441), the CRISPR RGEN tool program (http://www.rgenome.net/) (accessed on 10 March 2019) was used to select three target sites in the *PCP* sequence (Appendix A). Off-target analysis of the target sequences was performed on a website (http://www.rgenome.net/cas-offinder/) (accessed on 10 March 2019) [58]. The three gRNA targets were cloned into the pYLCRISPR/Cas9 vector and transformed into maize B104. We sequenced the predicted target sites in T1 (leaf) and T2 (pollen) progenies to validate homozygous *pcp* knock-out lines.

### 4.5. Scanning Electron Microscopy (SEM) of Pollen Grains

SEM was performed in accordance with previously described methods [22]. Briefly, freshly collected maize pollen grains were fixed in 2.5% glutaraldehyde (in 0.2 M PBS, pH 7.4) at 4 °C for 2 h. The pollen grains were then dehydrated in a series of increasing ethanol concentrations (30%, 50%, 70%, 80%, 90% and 100%) for 10 min each, and finally in tert-butanol for 15 min. The dehydrated pollen grains were left to air dry overnight at room temperature and coated with gold in the ion sputtering instrument (JFC-1600 AUTO COATER, JEOL, Japan) for 3 min (6 times, 30 s each). The morphology of the pollen grains was visualized using field emission SEM (JSM-7800F, JEOL, Japan).

### 4.6. Transmission Electron Microscopy (TEM) of Pollen Grains

TEM of maize pollen ultrastructure was performed in accordance with previously described methods [24] with a slight modification. Freshly collected pollen grains were fixed at 4 °C in 2.5% glutaraldehyde (in 0.1 M PBS, pH 7.4) overnight and then in 1% osmic acid (0.1 M PBS, pH 7.4) for 6 h. After fixation, the samples were washed several times in PBS and dehydrated in a series of increasing ethanol concentrations (30, 50, 70, and 90% for 15 min each) and then in acetone twice (for 15 min each). The dehydrated samples were embedded in a mixture of acetone and Epon812 resin (1:1, 1:3, for 30 min each; Sigma-Aldrich) and then in pure Epon812 resin for 2 h. The embedded blocks were polymerized at 37 °C and 45 °C for 12 h each, followed by 60 °C for 48 h. Sections were cut using the Leica EM UC7 ultramicrotome with a diamond knife. After staining with uranyl acetate and lead citrate, the sections were observed using a HT7700 microscope (Hitachi, Ibaraki, Japan).

### 4.7. Pollen Viability and Germination In Vitro

Pollen viability was examined with 2,3,5-triphenyltetrazolium chloride (TTC) staining [59] and I_2_-KI staining methods [60]. Pollen grains that were stained dark red or orange after 2 h of staining in 0.5% TTC solution were considered viable. Pollen grains that were stained blue after 5 min of staining in 0.5% I_2_-KI solution were also counted as viable. For in vitro germination assays, about 5 mg of fresh pollen grains was incubated in the dark at 25 °C in 5 cm × 1.5 cm Petri dishes with 1.5 mL of liquid medium containing 15% (*w*/*v*) sucrose, 10 ppm H_3_BO_3_, 100 ppm Ca(NO_3_)_2_, 37.5 ppm lysine and 0.05 ppm glutamic acid [61], with or without 10 μM E64 (MedChemExpress, Shanghai, China). E-64 is an irreversible specific CP inhibitor without affecting cysteine residues of other enzymes [62]. After 1–6 h of incubation, the germinated grains were counted. A pollen grain was considered germinated when its tube length reached pollen diameter [61,63].

All viability and germination tests were conducted in three biological replications. About 50 pollen grains in five randomly selected visual fields were counted in each replicate. The pollen staining and germination were observed under a light microscope (Eclipse Ni-U, Nikon, Tokyo, Japan).

### 4.8. Physiological and Biochemical Assays

Relative water content (RWC) of maize leaves was calculated using the following equation: RWC = (fresh weight − dry weight)/(turgid weight − dry weight)·100%. Malondialdehyde (MDA), usually used as an indicator of membrane lipid peroxidation [64], was extracted with 5% (*w*/*v*) trichloroacetic acid and quantified using a commercial kit (Art. BC0020, Solarbio, Beijing, China). Glutathione (GSH) is the most important sulfhydryl antioxidant in cells. GSH reacts with 5,5′-dithio-bis-2-nitrobenzoic acid (DTNB) to a yellow derivative, measurable at 412 nm [65]. GSH was determined via spectrophotometric assay at 412 nm using a commercial kit (BC1170, Solarbio, Beijing, China).

Maize leaf samples were homogenized on ice in the extraction buffer (0.10 g/mL) containing 50 mM PBS, pH 7.0, 1 mM EDTA and 1% polyvinylpyrrolidone. The homogenate was centrifuged (15,000× *g*, 15 min, 4 °C) and the supernatant was used for enzyme assays. All antioxidant enzymatic activities (SOD, POD, APX, GR, DHAR, MDHAR) and substance content (AsA, O_2_^−^) were measured using commercially available assay kits (Solarbio, Beijing, China), following the manufacturer’s manual. H_2_O_2_ were detected via 3,3-diaminobenzidine (DAB) staining (1 mg/mL in 10 mM Na_2_HPO_4_ solution) [66]. The dyed leaves were decolorized in 95% ethanol and were photographed.

### 4.9. Preparation of Paraffin Section of Root

WT, *pcp* mutants (*KO*) and *OE* seeds were germinated in the dark at 27 °C until the embryonic roots grew to approximately 6 cm. Subsequently, the 3–4 cm parts away from the root tip were sampled and fixed in 70% FAA solution for 48 h. Paraffin sections of the maize root tissues were made as follows: the root samples were immersed in 30% ethanol overnight, followed by 50% ethanol for 8 h. Then, the samples underwent dehydration, wax dipping and embedding. The trimmed wax blocks were sectioned on a paraffin microtome to a thickness of 5 μm. The sections were rehydrated successively and then stained with Plant Safranin Staining Solution and Green Staining Kit (Servicebio, Wuhan, China). The sections were examined using a microscope and scanned with a Pannoramic MIDI scanner (3DHISTECH, Budapest, Hungary), and images were acquired and analyzed.

### 4.10. RT-qPCR

Total RNA in maize tissues was extracted using RNA-Solv^®^ Reagent (Omega Bio-Tek, Norcross, GA, USA), and a 2 μg RNA sample was used to synthesize the first-strand cDNA via the 5×All-In-One MasterMix with AccuRT Genomic DNA Removal Kit (Applied Biological Materials Inc., Richmond, BC, Canada). The gene-specific primers of *PCP* were designed using the Primer Premier 5.0 software (http://www.premierbiosoft.com/) (accessed on 1 June 2019) and commercially synthesized (Biomed Cooperation, Beijing, China) as follows: *PCP*, 5′-AAGAAGCGGGCCAACGTATC-3′ and 5′-CCCTGTCGTGATCTTGGTGA-3′.

The reaction system of RT-qPCR was made using the kit of Hieff^®^ qPCR SYBR Green Master Mix (YEASEN, Shanghai, China) and performed in the Real-Time PCR Instrument Thermal Cycling Block (StepOnePlus^TM^, Applied Bio-systems, Waltham, MA, USA). The PCR conditions contained an initial denaturation step at 95 °C for 5 min, followed by 40 cycles at 95 °C for 10 s and 60 °C for 30 s. The quantification method 2^−ΔΔCT^ was used to assess the relative expression level of *PCP* after normalization based on *ZmUBI* expression [67]. Data were represented as relative expression (mean ± SD) from three biological replicates.

### 4.11. Statistical Analysis

Each experiment was performed in at least three biological replicates. Each biological replicate was performed using at least ten plants. Analysis of variance (ANOVA) was carried out via GraphPad Prism 8.0 software (San Diego, CA, USA). The significant differences (*p* < 0.05) were tested with one- or two-way ANOVA.

## 5. Conclusions

The present study highlights the crucial role of PCP in both pollen development and drought tolerance in maize. Our results suggest that while overexpressed *PCP* may negatively impact pollen germination, it positively enhances maize drought tolerance by regulating root structure remodeling. These findings offer new insights into the function of papain-like CP in maize and facilitate the development of drought-tolerant maize germplasm resources.

## Figures and Tables

**Figure 1 ijms-24-07406-f001:**
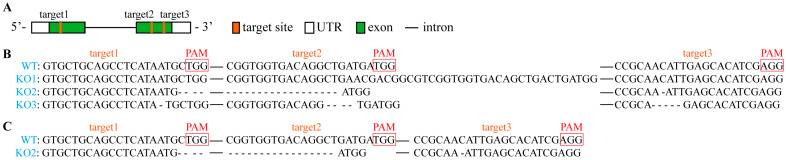
Knockout of *PCP* using CRISPR/Cas9 system. (**A**) The gene structure of *PCP* and three target sites (orange boxes) in the first and second exons of *PCP* for CRISPR/Cas9 editing. (**B**) Sequences of leaves in three homozygous knockout lines (T2) with deletions in target sites that truncated the *PCP* ORF (*KO1*, *KO2* and *KO3*). (**C**) Sequences of pollen in *KO2* line (T3). The sequence of the wild type is shown at the top. Target sites and protospacer-adjacent motif (PAM) sequences are highlighted in red, and the deletions are indicated by dashes. The sequence gap length is shown in solid lines.

**Figure 2 ijms-24-07406-f002:**
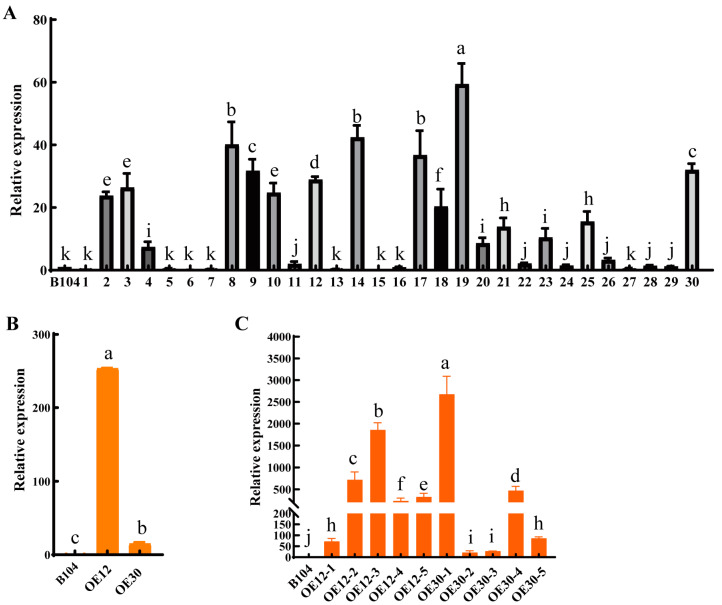
RT-qPCR analysis of the relative expression levels of *PCP* in overexpression plants. (**A**) T0 leaf at three leaf stage under laboratory conditions. (**B**) T1 leaf at anthesis stage under field conditions. (**C**) T2 pollen at anthesis stage under field conditions. Data represent means ± SD of three biological replicates. Significant differences in expression levels are indicated with different letters (*p* < 0.05, one-way ANOVA).

**Figure 3 ijms-24-07406-f003:**
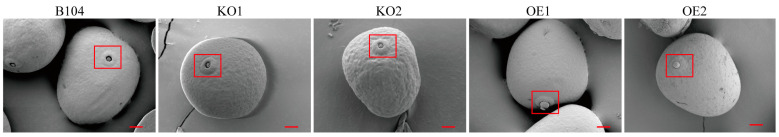
Representative SEM micrographs of pollen grains from WT and transgenic plants. The view highlights the aperture (inside the red box) of a pollen grain, showing the presence of a convex operculum covering the pore of *OE* pollen grain. Scale bar: 10 μm.

**Figure 4 ijms-24-07406-f004:**
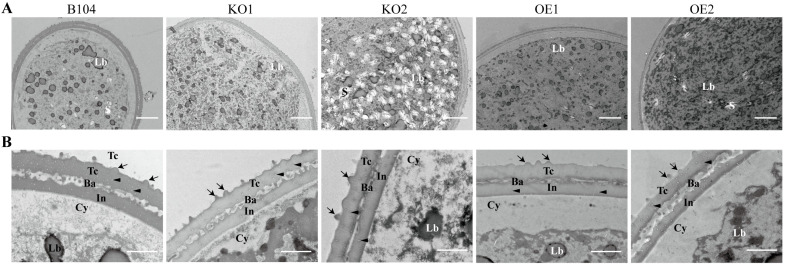
Representative TEM micrographs of pollen grains from WT and transgenic plants. (**A**) a pollen grain. Scale bar: 5 μm; (**B**) part of a pollen wall. Scale bar: 1 μm. TC: tectum; Ba: Bacula; In: Intine; Cy: Cytoplasm; S: Starch granules; Lb: Lipid body. Arrowheads indicate speckles present in the tectum, whereas arrows indicate the surface attachment of pollen exine.

**Figure 5 ijms-24-07406-f005:**
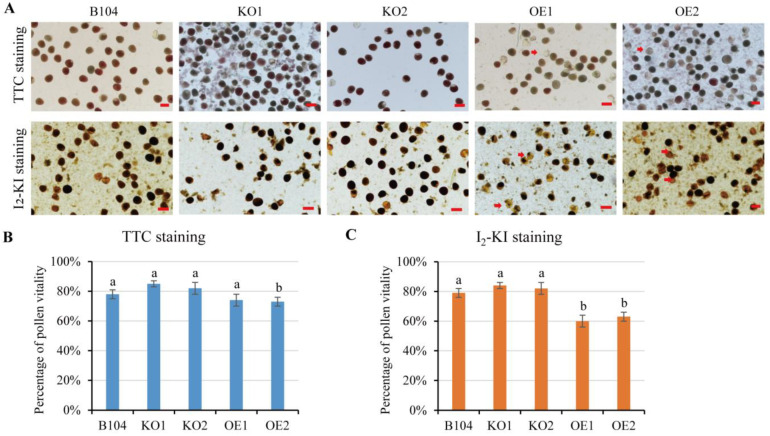
Evaluation of pollen vitality in vitro via TTC or I_2_-KI staining. (**A**) Microscopic observation of pollen via TTC or I_2_-KI staining. (**B**) Percentage of pollen vitality via TTC staining. (**C**) Percentage of pollen vitality by I_2_-KI staining. Scale bar: 100 μm. Data represent means ± SD of three biological replicates. Significant differences are indicated with different letters (*p* < 0.05, one-way ANOVA).

**Figure 6 ijms-24-07406-f006:**
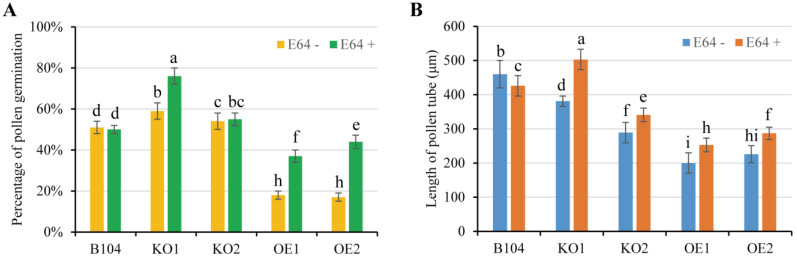
Comparison of pollen germination in vitro between WT and transgenic lines after 4 h incubation. (**A**) Percentage of pollen germination. (**B**) Length of pollen tube. Data represent means ± SD of three biological replicates. Significant differences are indicated with different letters (*p* < 0.05, two-way ANOVA).

**Figure 7 ijms-24-07406-f007:**
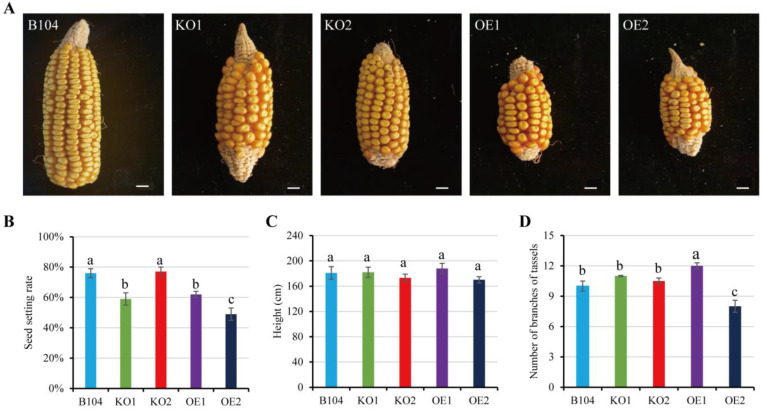
Comparison of plant height, branch number of tassels and kernel setting rate among WT and transgenic maize. (**A**) Representative mature female ears. Scale bar: 1 cm. (**B**) Seed setting rate. (**C**) Plant height. (**D**) Number of branches of tassels. Data represent means ± SD of three biological replicates. Significant differences are indicated with different letters (*p* < 0.05, one-way ANOVA, Duncan’s multiple range test).

**Figure 8 ijms-24-07406-f008:**
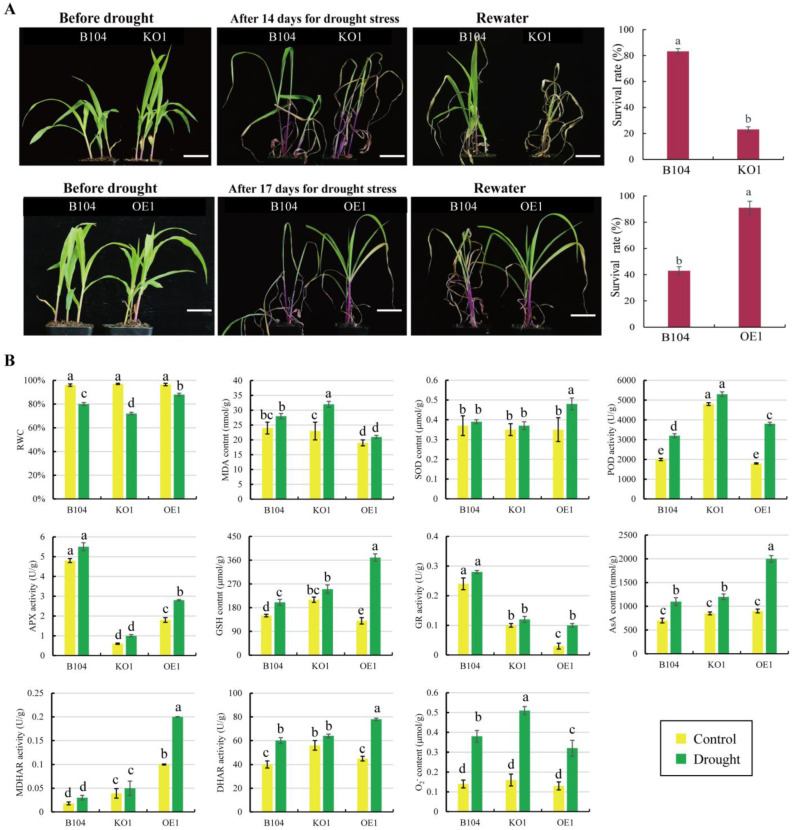
WT, *KO* and *OE* seedlings were examined for resistance to drought stress. (**A**) Phenotypes of seedlings under normal watering (control), drought and drought/rehydration conditions. Scale bar: 10 cm. (**B**) Physiological indexes of WT, *KO1* and *OE1* seedlings under drought conditions. RWC, relative water content; MDA, malondialdehyde; SOD, superoxide dismutase; POD, peroxidase; APX, ascorbate peroxidase; GSH, glutathione; GR, glutathione reductase; AsA, ascorbic acid; MDHAR, monodehydroascorbate reductase; DHAR, dehydroascorbate reductase; O_2_^−^, superoxide anion. Data represent means ± SD of three biological replicates. Significant differences are indicated with different letters (*p* < 0.05, two-way ANOVA).

**Figure 9 ijms-24-07406-f009:**
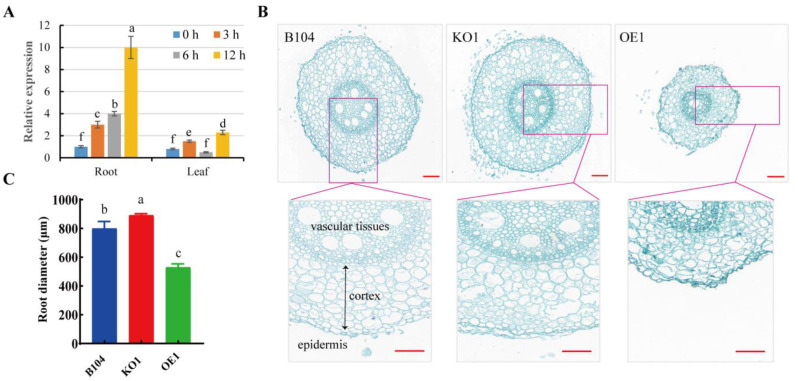
Comparison of *PCP* expression levels and root microscopic structure among WT and transgenic lines. (**A**) Relative expression levels of *PCP* in root and leaf of B104 under osmotic stress. (**B**) Cross section of root. Root sections were sampled at 30–40 mm from tips. Scale bar: 100 μm. (**C**) Diameter of roots. Data represent means ± SD of three biological replicates. Significant differences are indicated with different letters (*p* < 0.05, one-way ANOVA, Duncan’s multiple range test).

## Data Availability

Not applicable.

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
