# Peer review of "Modifying the Expression of Cysteine Protease Gene PCP Affects Pollen Development, Germination and Plant Drought Tolerance in Maize"

_ijms, 2023, doi:10.3390/ijms24087406_

Round 1

Reviewer 1 Report

The research presented in this paper is very interesting.

Author Response

Thank you very much for taking time to review our paper and giving us a positive evaluation. The submission has been substantially revised after incorporating all the comments. The manuscript has been copyedited by a professional language editing service to avoid the grammar and writing mistakes.

Reviewer 2 Report

The results are novel in comparison with previously published papers, are scientifically sound and relevant to the field. The article is suitable for publishing after minor spelling and English language are checked.

Author Response

Thank you very much for your valuable comments. The submission has been substantially revised after incorporating all the comments. The manuscript has been copyedited by a professional language editing service to avoid the grammar and writing mistakes. The changes made in the revision are highlighted in blue.

Reviewer 3 Report

Dear Authors,

I have read your manuscript with an interesting title: "Modification of PCP cysteine protease gene expression affects pollen development, germination and drought tolerance of maize plants". However, the paper has several flaws and shortcomings that should be eliminated before submitting the paper for publication.

Specific comments:

1.      The abstract should be structured. It should also be shortened.

2.      Introduction should end with a clear purpose of the paper or an alternative research hypothesis to the null hypothesis, rather than presenting the final conclusion.

3.      The research results are interestingly presented, but they should be presented in a clear way that readers can understand.

4.      Since an ANOVA analysis of variance was calculated, why were the significant differences not marked in the figures with letter designations, which would have made it possible to distinguish objects that were significantly different, but also homogeneous in terms of the studied characteristics (Figures: 2, 5-9).

5.      There are too few recent items in the Discussion to compare and discuss the results obtained.

6.      Conclusions should be generalizing and summarizing, not repeating the results. One conclusion at least should be towards the future.

7.      The research methodology is well presented.

Author Response

Thank you very much for your detailed and valuable comments. The submission has been substantially revised after incorporating all the comments. The manuscript has been copyedited by a professional language editing service to avoid the grammar and writing mistakes. The changes made in the revision are highlighted in blue. Below are our point-to-point responses to the comments.

Specific comments:

  1. The abstract should be structured. It should also be shortened.

Response: The abstract has been revised to be more concise and organized in the revision.

  1. Introduction should end with a clear purpose of the paper or an alternative research hypothesis to the null hypothesis, rather than presenting the final conclusion.

Response: The last paragraph of Introduction has been revised.

  1. The research results are interestingly presented, but they should be presented in a clear way that readers can understand.

Response: The results have been reorganized and modified in the revision. Besides, we added the results of ROS in WT, KO and OE plants in response to drought (Figure 8B, Figure S5).

  1. Since an ANOVA analysis of variance was calculated, why were the significant differences not marked in the figures with letter designations, which would have made it possible to distinguish objects that were significantly different, but also homogeneous in terms of the studied characteristics (Figures: 2, 5-9).

Response: The significant differences in Figures 2, 5-9 have been denoted with different letters, and the figure captions have been revised accordingly. Section 4.11, "Statistical Analysis," has also been revised.

  1. There are too few recent items in the Discussion to compare and discuss the results obtained.

Response: We have added several recent relevant references about CPs, pollen structure and root traits in the Discussion section.

  1. Conclusions should be generalizing and summarizing, not repeating the results. One conclusion at least should be towards the future.

Response: The conclusions have been revised in the revision to remove any duplicate content with the Results section.

  1. The research methodology is well presented.

Response: The RT-qPCR procedure and the methods of drought stress with PEG have been added in the Materials and Methods section.

Reviewer 4 Report

In this manuscript, Li and collaborators investigate the function of maize PCP protein on pollen morphology, viability and germination, as well as in the tolerance to drought. To this end, the authors created and analysed pcp knock-out (KO) and PCP overexpression lines (OE). The results obtained suggest that maize PCP OE impairs pollen germination and enhances drought tolerance likely by remodelling root structure.

In the opinion of this reviewer, this work is relevant because it contributes to shed light on the role of PCP proteins in pollen germination, tube growth and in the plant response to abiotic stress. English requires a revision as there are some typos and grammatical errors (see for instance: line 70 “No similar PCP was not identified in the pollen surface…”; line 76 “conservative” should be “conserved”; line 111 “homogonous” should be “homogenous”; line 124 “screened” should be “selected”; line 136 “expression level of PCP in leaves plant 2,…” should be “expression level of PCP in leaves of plants 2,…”).

Furthermore, there are several aspects of the work that need to be clarified in order to improve the understanding of the results and the main conclusions . See my comments below.

-        Section 2.1. The last two paragraphs are confusing. First, it is stated that DNA was extracted from KO2 mutant pollen to study the expression of the PCP gene. However it should be RNA and not DNA if PCP expression was investigated. Besides, no figure is shown to support the statement “The sequencing results of PCR products contained only single peaks, indicating that pcp was homozygous in the pollen of both mutants”. What does it means “both mutants”? KO2 is just one mutant. Along these lines, in the last paragraph of this section, it can be read “After measuring pcp expression via quantitative reverse transcription PCR (qRT-PCR) expression, three homozygous mutants were screened for further analysis, designated as KO1, KO2, and KO3 throughout”. Again, no figure is shown to support this statement and PCP is a gene and should be written in capital letters. qRT-PCR should be RT-qPCR.

 Section 2.2. I find the end of this section confusing. Were five OE plants derived from T0 12 (5 plants) and 30 (5 plants) finally selected for the rest of the studies in this work? I suggest rephrasing the last two sentences of this section for a better understanding of what it was finally done. Furthermore, can the authors give an explanation for the large differences in PCP expression levels in leaves of T0 and T1 OE12 plants, shown in figure 2A and B?

Section 2.3. Speckles in the tectum on Figure 4 should be indicated. In this figure, I suggest changing the colours of the letters, since yellow letters in a white background are difficult to read. Besides, surface attachment of pollen exine should be indicated in Figure 4.

Section 2.4. It can be read “the germination rate and tube growth of OE pollen grains was increased by adding E64 to a certain extent, especially after 4 h germination” A reference to Figure S1 is missing. Furthermore, it is stated later “the germination rate and tube growth between WT, KO1, and KO2 pollen had no significant differences, regardless of adding E64 or not (Figure 6, Figure S1), thus pcp mutants had no obvious effect on pollen germination in vitro” However, the percentage of pollen germination is significantly higher in KO1+E64 plants than in B104 (Figure 6A) and length of pollen tube is significantly reduced in KO2 +E64 or -E64 compared with B104 (Figure 6B). Moreover, mutants have not effect. Mutations are the ones that have effect (if any). This should be corrected.

Section 2.5. Explain the phenomenon of “retrogression” in OE2 plants (line 201). Besides, it can be read “The kernel setting rate of OE plants decreased and no significant difference between KO and WT plants was observed”. However, seed setting rate was significantly lower in KO1 than in B104 (Figure 7B).

Section 2.6. The results obtained with SOD, APX, GR, AsA and DHAR (Figure 8B) are not commented in the text of this section. Cortex cells should be indicated in Figure 9B. In the caption of this figure it can be read “after water loss in vitro”. However, in Methods, only soil drought treatment is reported.

 RT-qPCR procedure is not described in Methods.

Measurements of ROS such as H2O2 and O2- in WT, KO and OE plants in response to drought stress should be performed in order to support that antioxidant activities of enzymes in OE plants are higher than in WT leaves.

It can be read in Figures 5B, C, 6A, B, 7B-C and 9B captions “Error bars indicated the SE values of three biological replicates”. However, nothing is said about what the values represent. I guess it will be the average of several samples. This must be correctly described. In addition, it is indicated that three biological replicates have been used. Does this mean that only three plants per genotype were studied? If so, the results obtained would not be statistically valid. This must be clarified.

Author Response

Thank you very much for your detailed and valuable comments, which are very useful for improving the quality of our manuscript. The submission has been substantially revised after incorporating all the comments. The manuscript has been copyedited by a professional language editing service to avoid the grammar and writing mistakes. The changes made in the revision are highlighted in blue. Below are our point-to-point responses to the comments.

Furthermore, there are several aspects of the work that need to be clarified in order to improve the understanding of the results and the main conclusions . See my comments below.

-        Section 2.1. The last two paragraphs are confusing. First, it is stated that DNA was extracted from KO2 mutant pollen to study the expression of the PCP gene. However, it should be RNA and not DNA if PCP expression was investigated. Besides, no figure is shown to support the statement “The sequencing results of PCR products contained only single peaks, indicating that pcp was homozygous in the pollen of both mutants”. What does it means “both mutants”? KO2 is just one mutant. Along these lines, in the last paragraph of this section, it can be read “After measuring pcp expression via quantitative reverse transcription PCR (qRT-PCR) expression, three homozygous mutants were screened for further analysis, designated as KO1, KO2, and KO3 throughout”. Again, no figure is shown to support this statement and PCP is a gene and should be written in capital letters. qRT-PCR should be RT-qPCR.

Response: According to your advice, we have revised the last two paragraphs in the Section 2.1, and provided the sequencing results and the expression of the PCP gene in pollen (Figure S1, Figure S2). Also, we have made the following changes: "DNA" has been revised to "RNA," and "qRT-PCR" has been changed to "RT-qPCR.".

 Section 2.2. I find the end of this section confusing. Were five OE plants derived from T0 12 (5 plants) and 30 (5 plants) finally selected for the rest of the studies in this work? I suggest rephrasing the last two sentences of this section for a better understanding of what it was finally done. Furthermore, can the authors give an explanation for the large differences in PCP expression levels in leaves of T0 and T1 OE12 plants, shown in figure 2A and B?

Response: In Section 2.2, we have made the following revisions:

1) Added some content and corrected errors in the paragraph.

2) Clarified that the difference in PCP expression levels in the leaves of T0 and T1 OE12 plants was possibly due to differences in difference in plant growth environments.

3) Specified that the leaves of T0 were collected at the three-leaf stage under laboratory conditions, while the leaves of T1 were obtained at the anthesis stage under field conditions.

4) Changed "Five OE plants" to "PCP-OE-12 and PCP-OE-30 lines" and designated them as OE1 and OE2 for further analysis.

Section 2.3. Speckles in the tectum on Figure 4 should be indicated. In this figure, I suggest changing the colours of the letters, since yellow letters in a white background are difficult to read. Besides, surface attachment of pollen exine should be indicated in Figure 4.

Response: In Section 2.3, we have indicated the speckles in the tectum and the surface attachment of the pollen exine, as shown in Figure 4. We have made changes to the colour of letters in Figure 4 for better clarity.

Section 2.4. It can be read “the germination rate and tube growth of OE pollen grains was increased by adding E64 to a certain extent, especially after 4 h germination” A reference to Figure S1 is missing. Furthermore, it is stated later “the germination rate and tube growth between WT, KO1, and KO2 pollen had no significant differences, regardless of adding E64 or not (Figure 6, Figure S1), thus pcp mutants had no obvious effect on pollen germination in vitro” However, the percentage of pollen germination is significantly higher in KO1+E64 plants than in B104 (Figure 6A) and length of pollen tube is significantly reduced in KO2 +E64 or -E64 compared with B104 (Figure 6B). Moreover, mutants have not effect. Mutations are the ones that have effect (if any). This should be corrected.

Response: In Section 2.4, we have revised the description of the results on pollen germination in vitro between the WT and transgenic lines after a 4-hour incubation period.

Section 2.5. Explain the phenomenon of “retrogression” in OE2 plants (line 201). Besides, it can be read “The kernel setting rate of OE plants decreased and no significant difference between KO and WT plants was observed”. However, seed setting rate was significantly lower in KO1 than in B104 (Figure 7B).

Response: In Section 2.5, we have removed the sentence containing the word “retrogression” to avoid misunderstanding. We have also corrected the description of the results on kernel setting rates between the WT and transgenic lines.

Section 2.6. The results obtained with SOD, APX, GR, AsA and DHAR (Figure 8B) are not commented in the text of this section. Cortex cells should be indicated in Figure 9B. In the caption of this figure it can be read “after water loss in vitro”. However, in Methods, only soil drought treatment is reported.

Response: In Section 2.6, we have made the following revisions:

1) Added the results of SOD, APX, GR, AsA, and DHAR activities (shown in Figure 8B) to the revised section.

2) Indicated the cortex cells in Figure 9B.

3) Corrected the caption of Figure 9.

4) Added the osmotic stress treatment method to the Materials and Methods section.

 RT-qPCR procedure is not described in Methods.

Response: RT-qPCR procedure has been added in the Materials and Methods section.

Measurements of ROS such as H2O2 and O2- in WT, KO and OE plants in response to drought stress should be performed in order to support that antioxidant activities of enzymes in OE plants are higher than in WT leaves.

Response: The measurements of ROS such as H2O2 and O2- in WT, KO and OE plants exposed to drought have been performed, and the results have been added in the Section 2.6 (Figure 8B, Figure S5).

It can be read in Figures 5B, C, 6A, B, 7B-C and 9B captions “Error bars indicated the SE values of three biological replicates”. However, nothing is said about what the values represent. I guess it will be the average of several samples. This must be correctly described. In addition, it is indicated that three biological replicates have been used. Does this mean that only three plants per genotype were studied? If so, the results obtained would not be statistically valid. This must be clarified.

Response: We have made corrections to the Statistical Analysis in the Materials and Methods section and captions of Figures 5B, C, 6A, B, 7B, C, and 9B regarding their statistical analyses. The revised captions read as follows:

" Data represent means ± SD of three biological replicates. Significant differences in expression levels are indicated with different letters (P < 0.05, one-way ANOVA)."

Round 2

Reviewer 4 Report

Authors have successfully addressed most of the concerns and suggestions raised in my previous revision of the manuscript. Thus, the changes made have strengthened the manuscript which has been substantially improved. Nonetheless, in my opinion, there are still some points that should be addressed for the paper to be acceptable for publication.

First, in section 2.3, it can be read (lines 167-169) “On the other hand, the KO mutant had a small number of speckles, whereas the overexpression strain did not have any”. However, this is contradictory with the two arrowheads shown in the OE2 picture.

Section 2.4. Figure 6B. I wonder why length of pollen tube in KO2 is significantly reduced compared with the WT. Since PCP functions as a negative regulator of pollen growth, wouldn´t an increase rather than a decrease in pollen tube length be expected? Why does KO2 pollen length is significantly lower than that of KO1 plants?

Authors have changed the text of section 2.5 according to my suggestions. However, I still find some inconsistencies in the results depicted in Figures 7B and D. In the first one, authors should explain why seed setting is very similar in KO1 and OE1, and significantly lower in KO1 than in the WT. Moreover, how to explain the different results in the number of branches of tassel of OE1 and OE2 plants? In other words, why are OE1 and OE2 values significantly higher and lower respectively, than those of the WT?

In section 2.6 it can be read (lines 235-237) “These results indicate that OE plants possess strong anti-oxidation machinery that effectively scavenges the accumulation of intracellular ROS”. However, this statement does not match with the results shown in the graph of superoxide anion of Figure 8B where there are not significant differences in O2- levels between OE1 and the WT in response to drought stress. Moreover, superoxide anion and DAB staining results in OE1 plants (Figure S5) seems contradictories, since DAB staining reveals decreased levels of H2O2 in OE1 compared with the WT.

A “b” letter in the OE1 green bar of the MDA graph should be removed.

Lines 436-437. It can be read “ROS were detected by 3,3-diamino-benzidine (DAB) staining (1 mg/mL in 10 mM Na2HPO4 solution”. It should be specified that the ROS detected is H2O2 rather than ROS in general.

Author Response

We sincerely appreciate you taking the time to go through our article twice. Your comments on the inaccuracies in the data and explanations of the results are very useful for improving our article. Your comments have also motivated us to refine our next research. With a lot of T3 transgenic seeds in hands, we will investigate the molecular mechanisms of PCP affecting pollen development, germination and tube growth this summer. We've done our best to respond to your questions. The revision's modifications are indicated in blue.

First, in section 2.3, it can be read (lines 167-169) “On the other hand, the KO mutant had a small number of speckles, whereas the overexpression strain did not have any”. However, this is contradictory with the two arrowheads shown in the OE2 picture.

Response: We double-checked the TEM pictures of all the transgenic maize pollen and noticed that the speckles were present in pollen walls of all WT, KO, and OE plants. The lack of visibility of the speckles in the KO2 photographs may be caused by variations in section preparation and image acquisition. In this revision, the KO2 picture in Figure 4B has been replaced, and the inaccurate description of the speckles in the text has been removed.

Section 2.4. Figure 6B. I wonder why length of pollen tube in KO2 is significantly reduced compared with the WT. Since PCP functions as a negative regulator of pollen growth, wouldn´t an increase rather than a decrease in pollen tube length be expected? Why does KO2 pollen length is significantly lower than that of KO1 plants?

Response: We have made the amendment to clarify any unclear findings descriptions in Section 2.4. Although the pollen tube in KO2 was shorter than that in WT initially, there was no visible difference in their final length after 24 h of in vitro incubation (data not shown). We had no idea about this. Our findings imply that PCP inhibits pollen germination by acting as a negative regulator. The individual differences between the two genotypes may be responsible for the variation in pollen tube length between KO1 and KO2 cultivated in vitro for 4 h.

Authors have changed the text of section 2.5 according to my suggestions. However, I still find some inconsistencies in the results depicted in Figures 7B and D. In the first one, authors should explain why seed setting is very similar in KO1 and OE1, and significantly lower in KO1 than in the WT. Moreover, how to explain the different results in the number of branches of tassel of OE1 and OE2 plants? In other words, why are OE1 and OE2 values significantly higher and lower respectively, than those of the WT?

Response: We tried to explain this in Section 2.5. Many variables, including genotype and environment, may have an impact on the difference in seed setting rate between WT and transgenic plants. The variation in the number of tassel branches between OE1 and OE2 may be a result of the two genotypes' unique characteristics. We largely paid attention to pollen features rather than other agronomic parameters

In section 2.6 it can be read (lines 235-237) “These results indicate that OE plants possess strong anti-oxidation machinery that effectively scavenges the accumulation of intracellular ROS”. However, this statement does not match with the results shown in the graph of superoxide anion of Figure 8B where there are not significant differences in O2levels between OE1 and the WT in response to drought stress. Moreover, superoxide anion and DAB staining results in OE1 plants (Figure S5) seems contradictories, since DAB staining reveals decreased levels of H2Oin OE1 compared with the WT.

Response: We have re-measured the content of O2- in the leaves of WT and transgenic plants. The results description has been updated in the version.

A “b” letter in the OE1 green bar of the MDA graph should be removed.

Response: The error in Figure 8 has been corrected.

Lines 436-437. It can be read “ROS were detected by 3,3-diamino-benzidine (DAB) staining (1 mg/mL in 10 mM Na2HPO4 solution”. It should be specified that the ROS detected is H2O2 rather than ROS in general.

Response: The “ROS” has been changed to “H2O2”.

Round 3

Reviewer 4 Report

Authors have successfully addressed my concerns although I have a couple of suggestions and a question. I recommend to comment in the discussion that the differences between the KO1 and KO2 genotypes may be responsible for the variation in pollen tube length between KO1 and KO2 cultivated in vitro for 4 h. Besides, I suggest to indicate that the variation in the number of tassel branches between OE1 and OE2 might be attributable to the different levels of expression of the PCP gene in the OE1 and OE2 lines. Finally, I wonder why the survival rate of WT B104 plants after drought exposure and rewatering, is so different in the graphs depicted in Figure 8A: 80% and 40%, respectively.

Author Response

According to your suggestion, we have added the sentences to discuss the variations among these transgenic mutants. It reads as below:

“In addition, in this study, KO2 pollen length was significantly lower than that of KO1 plants incubated in vitro for 4 h. The variance in pollen tube length between KO1 and KO2 may be caused by their different mutation locations (Figure 1) and PCP expression levels (Figure S2). Similar phenomena have been observed in maize [37] and Setaria viridis [38] KO mutants. For instance, internode lengths and numbers were clearly different in the ZmGA20ox3 gene KO mutants [37]. A single codon deletion mutant and two ANT1 orthologous frameshift mutants of Setaria viridis generated through CRISPR/Cas9 technology displayed noticeable variations in photosynthesis efficiency, growth rate, leaf size, and grain yield [38]. Moreover, the variation in the number of tassel branches between OE1 and OE2 plants might be attributable to the differential levels of expression of the PCP gene. Individual OE lines of maize (e.g., YIGE1-OE, ZmPHYC1-OE) showed differences to some extent in ear length, grain yield [39] and organ growth [40]. The molecular mechanisms underlying these phenotype differences are largely unclear.”

You correctly noted that it was absurd for the survival rate of B104 to have changed so widely following the same period of soil drought treatment. In our initial submission, we made it clear in the upper and lower panels of Figure 8A that the soil drought treatment lasted for 14 and 17 d, respectively. The stress time was incorrectly adjusted to 14 d throughout the language editing process. This issue has been fixed in both Figure 8A and the text.

Finally, we wish to thank you for your valuable comments and suggestions. Have a good day!